# Effects of menstrual cycle on cognitive function, cortisol, and metabolism after a single session of aerobic exercise

**Maryam Mirzaei Khalil Abadi, Mohammad Hemmatinafar, Maryam Koushkie Jahromi** [ID]*

Department of Sport Sciences, School of Education and Psychology, Shiraz University, Shiraz, Iran

* koushkie53@yahoo.com, mkoushki@rose.shirazu.ac.ir

**Data Availability Statement:** All relevant data are within the paper and its Supporting Information files.

## Abstract

### Aim

This study aimed to investigate the effects of the two pre-ovulatory and mid-luteal phases of the menstrual cycle on cognitive function, as well as possible mediators of metabolism and salivary cortisol, at rest and after an aerobic exercise session.

### Study design

Twelve active young unmarried women aged 22–30 years volunteered to participate in the study. The participants performed a 20-min exercise session on a cycle ergometer at 60–70% of their reserve heart rate twice, during the follicular (pre-ovulation: days 7–10) and luteal (mid-luteal: days 21–24) phases of the menstrual cycle. Saliva samples were collected to measure cortisol. Fat utilization, respiratory exchange ratio (RER), and energy expenditure (during exercise) were measured using a spiroergometer. Cognitive function was assessed using the Stroop test. Cognitive function and cortisol levels were measured before and after each exercise session.

### Results

The findings of this study indicated no significant differences in variables during the resting follicular and luteal phases. Cortisol levels and cognitive function were increased after exercise compared with before exercise in both the follicular and luteal phases. Cortisol and fat utilization after exercise were significantly higher in the follicular phase than in the luteal phase. There were no significant differences between the follicular and luteal phasesregarding the effects of exercise on cognitive function, energy expenditure, and RER.

### Conclusion

In general, the follicular and luteal phases of menstruation may not affect cognitive function in response to a single aerobic exercise session, although they change some metabolic factors and cortisol.

**Funding:** The author(s) received no specific funding for this work.

## Introduction

Recently, many women have participated in physical activities to maintain their health, and accordingly, special attention to their physiological and psychological demands and health has also increased [1,2]. Women have unique states during their menstrual cycles. Their reproductive hormone levels fluctuate significantly during the menstrual cycle. Female reproductive hormones (such as estrogen and progesterone) have various biological functions beyond reproduction [3,4]. Estrogens facilitate insulin secretion, regulate glucose availability, promote the utilization of lipids as the primary energy substrate, and influence whole-body metabolism through neurohormonal mechanisms [5]. It can regulate the release of other hormones, such as cortisol, which influences lipolysis and fat utilization [6]. Estrogen has neuroprotective effects and can alleviate verbal memory impairment caused by anticholinergics [7]. Estrogen positively affects the brain and cognition, whereas glucocorticoids, including cortisol, can negatively affect the brain and cognition [8]. However, there is evidence that progesterone has a largely antiestrogenic effect [9,10]. Circulating cortisol levels are lower, while circulating progesterone is higher during the luteal phase than during the follicular phase[11]. Serum fatty acids are also negatively associated with cognitive function [12]. In this regard, cortisol increases lipolysis and serum fatty acid levels, while estradiol increases lipolysis and decreases serum fatty acid levels by increasing cellular uptake. Previous studies considered two phases, follicular and luteal, while progesterone and estrogen are highly variable on different days of these two phases. During pre-ovulatory phase (days 7–10 of the menstrual cycle) compared with mid-luteal phase (days 21–24 of the menstrual cycle), estrogen is similarly high. In contrast, during days 21–24 of the mid-luteal phase, progesterone levels are significantly higher than in the pre-ovulatory phase due to the corpus luteum's activity [13]. Considering the contradictory effects of progesterone, cortisol, and lipolysis on cognitive function, it is necessary to evaluate the possible cognitive changes during the different days of follicular and luteal phases.

Physical activity may also influence cognitive function via the sympathoadrenal system, including norepinephrine (NE) and cortisol [14]. Greater arousal of the sympathetic system is associated with a more significant increase in norepinephrine (NE) levels and may lead to a reduction in accuracy following acute resistance exercise [15]. Exercise sessions increase serum fatty acids and catabolic hormones, including NE, cortisol, and epinephrine [16]. There are controversies regarding the effects of an exercise session with different durations and intensities on cognitive function [17]. However, no study was found on the effects of progesterone fluctuation phases during the pre-ovulatory phase (days 7–10 of the menstrual cycle) and mid-luteal phase (days 21–24 of the menstrual cycle) on cognitive function in response to exercise.

Considering the effects of exercise and sex hormones on lipolysis, cognitive function, and cortisol levels, this study aimed to assess whether the two pre-ovulatory and mid-luteal phases of the menstrual cycle have different effects on cognitive function at rest and in response to exercise through possible mechanisms of lipolysis and cortisol.

## Materials and methods

### Participants

This study was conducted at Shiraz University, one of the largest and most important universities in Iran, with over 20000 students. Participants were recruited through announcements in university dormitories and gymnasiums from January to April 2023. Twenty-six female postgraduate students volunteered to participate in the study, and 12 active young female students

aged 22–30 years who had a history of participating in sports training regularly at least two sessions a week in the past year and met the inclusion criteria were selected as study participants. All of them were unmarried and had no history of pregnancy or childbirth. The inclusion criteria were (a) an age range between 22 and 30, (b) a normal and regular menstrual cycle (the duration of the cycle ranges from 21 to 45 days, typically involving three to seven days of bleeding on average) [13], (c) physical and mental health to perform exercise(approved by a physician), (d) no metabolic or endocrine diseases (e.g. diabetes, thyroid gland diseases, insulin resistance, high blood pressure, polycystic ovarian syndrome(PCOs), etc.), (e) no high depression or anxiety, (f) no obesity (BMI of 30 kg/m2 or higher) [18], and absence of hormonal treatment (e.g. oral contraceptives). Exclusion criteria were (a) any type of menstrual abnormality or disease during the intervention that could affect the study variables and (b) no satisfaction with continued participation in the study. The Hospital Anxiety and Depression Scale (HADS) questionnaire was used to measure anxiety and depression before participating in the study. A score of 0–7 was considered normal [19]. Participants with a score of >7 were excluded from the study.

Among the 26 volunteers participating in the study, 15 were selected according to the inclusion criteria, and three were excluded according to the HAD score.

## Procedures

The Shiraz University ethics committee approved the study proposal and procedures(No.: IR. US.REC.1401.037). All participants signed a written informed consent form before participating in the study.

Measurements were performed from April to July 2023. The number of participants was estimated as 12 by G*power software (effect size f = 0.37, α error prob = 0.05, power(1-βprob) = 0.80, number of groups = 1, number of measurments = 4, correlation among measurments = 0.5). The possible effect size was estimated according to a publication that evaluated the effect of the menstrual cycle on sports performance [20].

Participants' weight and height were measured in standard positions. Body mass index (BMI) was calculated as weight (kg)/height (m)$^2$. To monitor the regularity and date of menstruation, participants were taught to record the exact date of the start and end of their menstrual cycle and bleeding, and their resting heart rate (after waking) using a Polar H10 Heart Rate Sensor(made in Kempele, Finland) for three months before the start of the study. The validity of estimating ovulation through heart rate has been approved [21]. A slight increase in resting heart rate, as well as recorded information considering the menstrual cycle, were considered for predicting follicular, luteal, and ovulation dates. The follicular (beginning from the first day of menses until ovulation) and luteal (beginning around day 15 of a 28-day cycle and ending with menstruation) phases of the participants were estimated and recorded by the researcher according to the information provided. All participants participated in two exercise sessions in the pre-ovulatory phase (days 7–10 of the menstrual cycle) and another in the mid-luteal phase (days 21–24 of the menstrual cycle) [22] in a crossover design to limit the effect of learning or adaptation. Thus, in the first and second sessions, 6 participants were in the follicular phase and 6 in the luteal phase. To control for the effects of the natural circadian rhythm of cortisol levels, exercise sessions, and measurements were performed between 12:00 and 14:00. Participants were asked not to exercise or drink alcohol for 24 hours, not to consume caffeine or food, and not to smoke for 3 hours before the exercise and measurement sessions[23]. Fuel utilization and energy expenditure were measured with the Metalyzer 3B (made in Germany) ergospirometry system during each exercise session. Cognitive function and cortisol levels were measured using the Stroop test before and immediately after each exercise session.

## Exercise protocol

This was a nonrandomized crossover study. Before starting the exercise, the seat of the cycle ergometer was adjusted for each participant, who was asked to pedal on the cycle ergometer (Ergoline Ergomedic 839 E; Monark, Germany) for 30 minutes, starting at 25 watts. The exercise protocol included 5 min of warm-up, 20 min of steady-state cycling at 60–70% heart rate reserve, and 5 min of cool-down cycling [24]. During the steady-state phase, the cycling speed was maintained at 65 rpm, and the workload was checked and adjusted every 2 min to maintain the cycling speed at an individually determined target HR. The target HR was estimated using the Karvonen formula as follows:

Target HR = heart rate reserve ([206–0.88 × age—resting HR]) × percentage of target intensity + resting HR).

Heart rate was continuously monitored using Metalyzer 3B.

Metalyzer 3B analyzed respiratory and metabolic parameters, including oxygen consumption (VO2), carbon dioxide production (VCO2), respiratory exchange ratio (RER), and heart rate. To measure gas exchange, participants wore a comfortable and adjustable face mask during exercise sessions that fit securely and minimized discomfort during exercise.

## Cognitive task measurement

The validated Persian version of the Stroop Color and Word Test (SCWT) was used to assess cognitive performance [25]. The Stroop cognitive function test was administered shortly before the start of the exercise and 10 min after the exercise performance using the Stroop test software. This test includes four colors: green, yellow, red, and blue. In this way, participants were shown a total of 480 congruent words (the color of the word was the same as its script) and incongruent words (the color of the word was different from its script) and were asked to respond as quickly as possible [26]. The speed of their response was recorded in milliseconds (ms). They performed the test using the index fingers of both hands.

## Saliva sampling and assay

Participants were asked not to eat, drink (except water), smoke, or brush their teeth for 30 min before collecting saliva samples to ensure that external factors do not influence the saliva composition. To measure salivary cortisol, salivary samples were collected immediately before and 30 min after the two exercise sessions to evaluate cortisol levels in response to physical activity. The mouth was rinsed before sample collection to wash away contaminants and residual substances that might affect the sample quality. The tongue was then moved into the mouth for 2 min to stimulate saliva secretion, and saliva was poured into the tube. The collected saliva samples were immediately stored at −20˚C to preserve their components for further testing. The saliva samples were thawed completely and centrifuged at 3000 rpm for 15 min. This separates the clear liquid (supernatant) from any solid residue (pellet), allowing for more precise measurement of the saliva's components. Salivary cortisol levels were measured using a commercially available Salivette kit (Eliza, Diametra, Italy). Salivary cortisol concentrations were determined using a specific laboratory technique of electrochemiluminescence immunoassay. Cortisol levels were measured according to the manufacturer's instructions. The tests were performed three times for each sample to ensure accuracy and reliability. For cortisol, the intra-assay variability was ≤10% indicating good consistency when measuring the same sample in one session and the inter-assay variability was ≤8.3% showing reliability over time. The sensitivity of the kit to determine cortisol was 0.12 g/ml with a confidence limit of 95, emphasizing its precision in measuring cortisol.

## Statistical analyses

To analyze the results, after confirming the normal distribution of the data with the Shapiro–Wilk test, paired t-test and Analysis of Variance (ANOVA) with repeated measures were performed to compare the measurements. For paired comparisons, the Bonferroni test or paired t-test was used for paired values. Statistical analysis was performed using SPSS software (version 21).

# Results

Participants of the study included 12 collegiate active females (age:24.91±1.19 years, height:162.82±7.02 cm, weight:54.3±7.73 kg, and BMI:20.30±1.49 kg/m$^2$), with menstrual cycle duration of 28.08±1.62 days.

## Cortisol

Repeated measures ANOVA showed a significant difference between cortisol levels in the different phases of the menstrual cycle and exercise($F_{(1, 11)}$ = 13.387, P = 0.030). Paired-wise comparisons showed that at rest, cortisol level in the follicular phase(mean 5.24, 95% CI 3.38, 6.60) was non-significantly (mean difference-0.87, 95% CI -0.53, 2.27) higher than the luteal phase (mean 4.37, 95% CI 3.69, 5.05). In the follicular phase, exercise cortisol levels following exercise(mean 7.62, 95% CI 6.61, 8.62) increased significantly(mean difference -2.37, 95% CI -3.57, -1.17; p = 0.002) compared to before exercise(mean 5.24, 95% CI 3.88, 6.60). In luteal phase, also cortisol levels after exercise(mean 6.294, 95% CI 5.42, 7.16) increased significantly (mean difference -1.92, 95% CI -2.77, -1.06; p = 0.001) compared to before exercise(mean 4.37, 95% CI 3.69, 5.05).

Paired-wise comparisons indicated that cortisol levels after exercise in the follicular phase (mean 7.620, 95% CI 6.61, 7.62) were significantly higher (mean difference 1.42, 95% CI 0.70, 1.95; P = 0.001) than in the luteal phase (mean 6.294, 95% CI 5.42, 7.16). However, a comparison of changes (post-exercise–pre-exercise) using paired t-test indicated that cortisol changes during the follicular phase(mean change 2.37, 95% CI 1.43,3.35) were not (mean difference -4.55, 95% CI -1.75, 0.84; t = -0.789, p = 0.450) higher than in the luteal phase(mean change 1.92, 95% CI 1.25, 2.66) significantly (Fig 1). The effect size of cortisol changes due to exercise was greater in the follicular phase than in the luteal and this amount was greater than medium ($\eta 2p$ = 0.065).

## Metabolism

As shown in Fig 2, energy expenditure was not significantly different between the follicular and luteal phases (mean difference 28.70, 95% CI-36.97, 114.27; t = 1.156, p = 0.277). However, it was higher(non-significantly) in the follicular phase (mean 460.00, 95% CI 379.14, 544.99) than in the luteal phase (mean 421.30, 95% CI 373.70, 468.98), and the effect size was more than medium ($\eta 2p$ = 0.129).

There was a significant difference in fat utilization during exercise between the follicular and the luteal phases (mean difference 3.70, 95% CI 0.001, 0.977; t = 6.871, p<0.001). Fat utilization in the follicular phase (mean 18.80, 95% CI 16.80, 21.00) was significantly higher than that in the luteal phase (mean 15.10, 95% CI 13.60, 17.20) (Fig 3), and the effect size was large ($\eta 2p$ = 0.840).

The RER was not significantly different between the measurements (mean difference -0.001, 95% CI -0.02, 0.02, t = 0.087, p = 0.933). It was higher(non-significantly) in the follicular phase (mean 0.893, 95% CI 0.85, 0.93) than in the luteal phase (mean 0.892, 95% CI 0.85, 0.92) (Fig 4), while the effect size was small ($\eta 2p$ = 0.001).

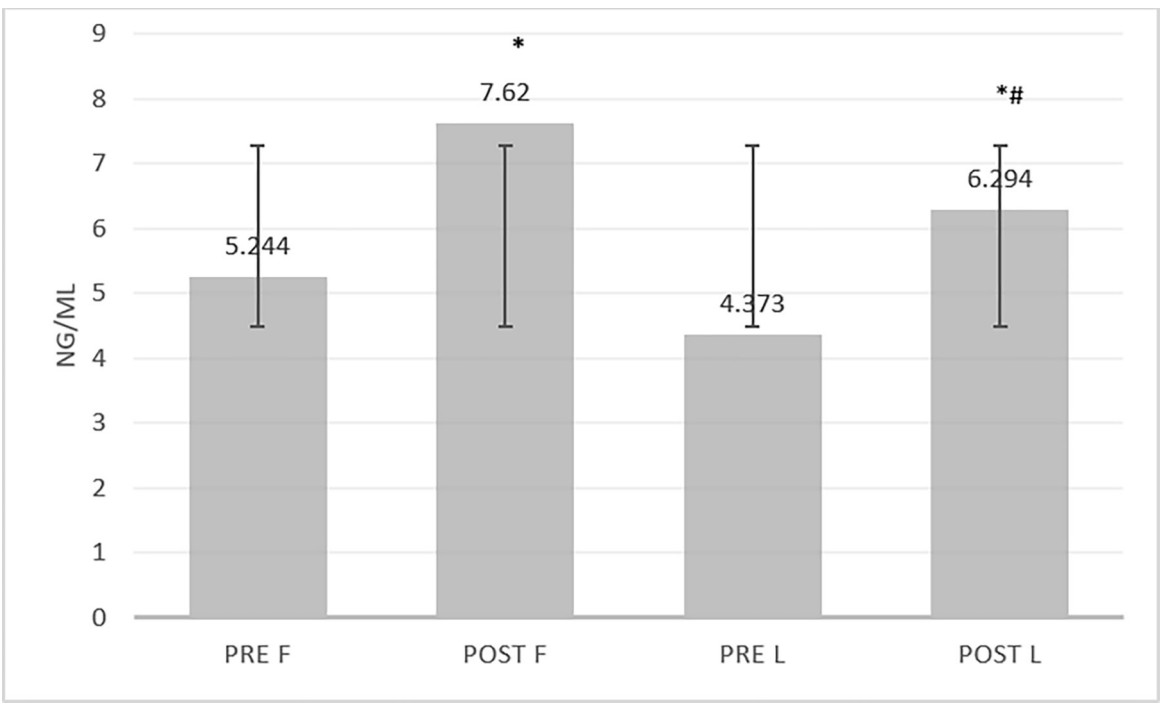

**Fig 1. Comparison of cortisol at four different times (pre-exercise follicular phase (PRE F), post-exercise follicular phase (POST F), pre-exercise luteal phase (PRE L), and post-exercise luteal phase (POST L)).** *significant difference with pre-exercise;. # significant difference with follicular phase.

## Cognitive function

There was a significant difference in the reaction time to congruent ($F(1, 11) = 8.332$, $P = 0.010$) and incongruent ($F(1, 11) = 12.299$, $P = 0.001$) colors in the different phases of menstruation.

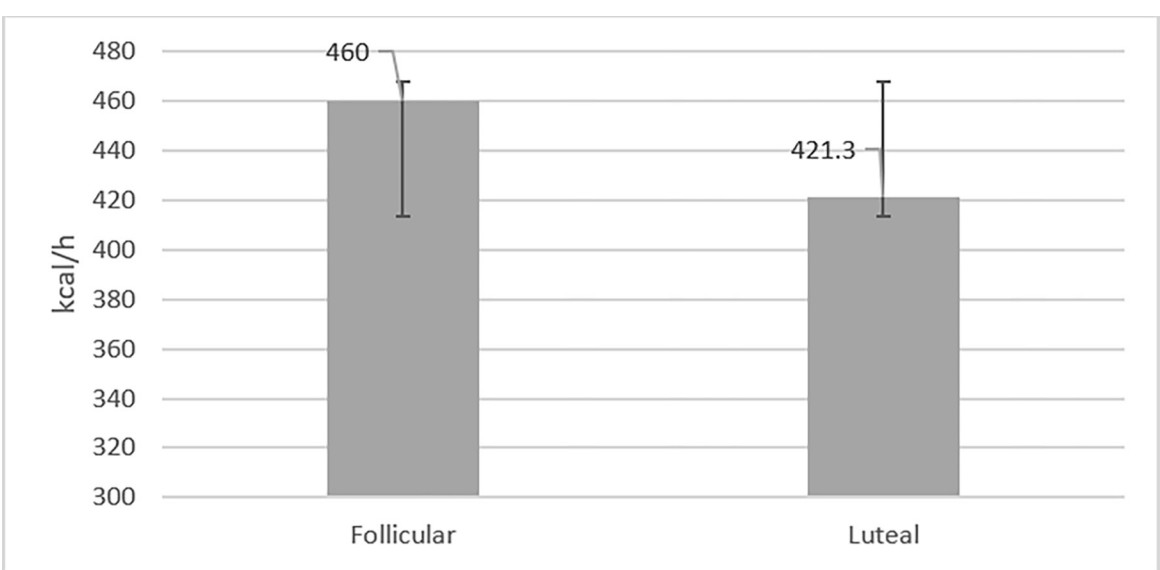

**Fig 2. Comparison of energy expenditure(EE) during exercise in follicular and luteal phases.**

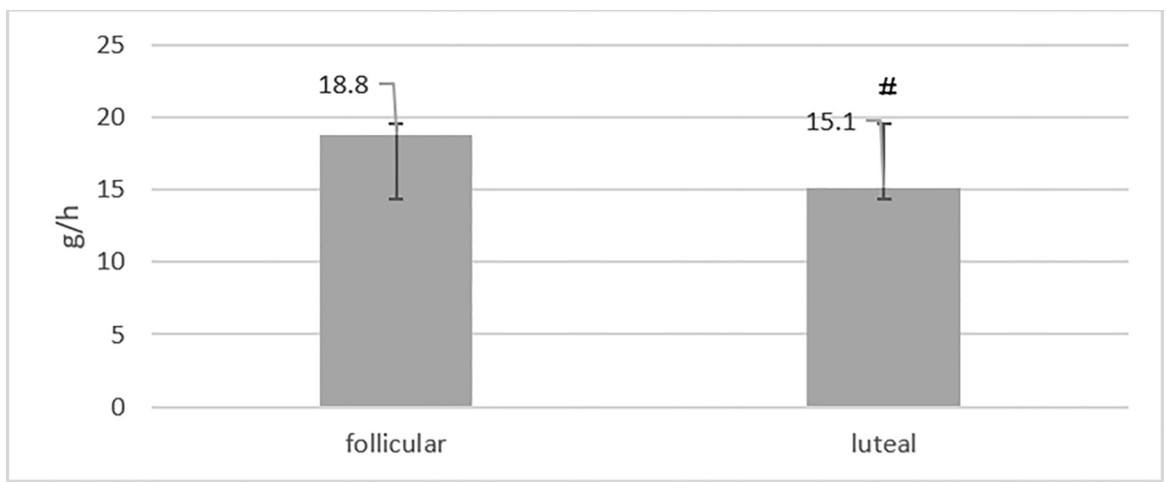

**Fig 3. Comparison of fat utilization during exercise in follicular and luteal phases. # significant difference with follicular phase.**

In the follicular phase, the reaction time to congruent colors decreased significantly after exercise (mean 664.8, 95% CI 621.49, 708.11) compared to before exercise (mean 700.5, 95% CI 654.39, 746.60) (mean difference 35.50, 95% CI 15.95, 55.45; P = 0.003). In addition, in the luteal phase, reaction time to congruent before exercise (mean 701.400, 95% CI 656.22, 745.57) was significantly higher than that after exercise (mean 656.900, 95% CI 605.26, 704.53) (mean difference 44.50, 90% CI 0.53, 88.46; p = 0.048). There was no significant difference in resting reaction time for congruent colors between the follicular (mean 700.5, 95% CI 654.39, 746.60) and luteal(mean 701.40, 95% CI 656.22, 745.57) phases(mean difference -0.90, 95% CI -52.46, 50.66; p = 0.969). Post-exercise comparisons of reaction time for congruent colors between follicular (mean 664.8, 95% CI 621.49, 708.11) and luteal(mean 656.90, 95% CI 629.26, 704.53) phases indicated that there was no significant difference between the follicular and luteal phases (mean difference 7.90, 95% CI -15.42, 31.22, p = 0.463). In addition, comparison of changes (post-exercise–pre-exercise) using t-test indicated that reaction time for congruent colors was non-significantly higher(mean difference 8.80, 95% CI -42.52, 60.12, t = 0.388,

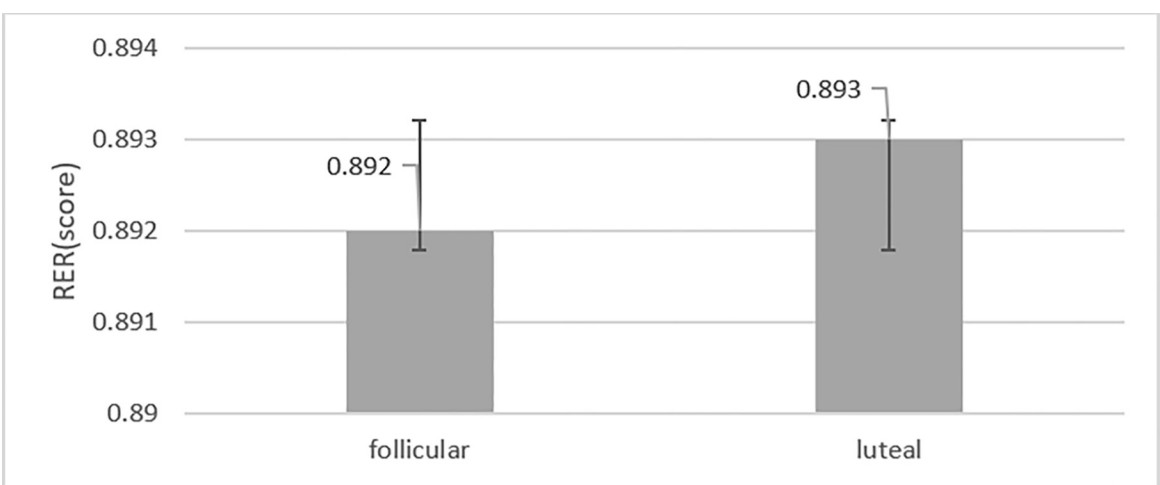

**Fig 4. Comparison of respiratory exchange ratio(RER) during exercise in follicular and luteal phases.**

p = 0.707) in the follicular phase(mean -35.70, 95% CI -49.99, -17.70) than in the luteal phase (mean -44.50, 95% CI -80.39, -9.01, t = 0.388, p = 0.707), while the effect size was small ($\eta$2p = 0.016).

At rest, there was no significant difference between the follicular and luteal reaction time to incongruent colors(mean difference -2.30, 95% CI -66.19, 61.59, p = 0.937). In the follicular phase, the reaction time to incongruent colors after exercise (mean 705.20, 95% CI 650.71, 759.68) decreased significantly compared to before exercise (mean 747.50, 95% CI 691.21, 803.78) (mean difference 42.30, 95% CI 23.67, 60.93, P = 0.001). In the luteal phase, the reaction time to incongruent colors after exercise decreased significantly (mean 692.50, 95% CI 650.38, 734.61) compared to that before exercise(mean 749.80, 95% CI 689.34, 810.25)(mean difference 57.30, 95% CI 13.58, 101.01, p = 0.016).

After exercise, paired comparisons indicated no significant difference in reaction time to the incongruent colors in the follicular phase(mean 705.20, 95% CI 650.71, 759.68) compared to the luteal phase(mean 692.50, 95% CI 650.38, 734.61) (mean difference 12.70, 95% CI -21.59, 46.99, p = 0.424). In addition, a comparison of changes (post-exercise–pre-exercise) using t-test indicated that reaction time for incongruent colors was not statistically significant (mean difference -15.00, 95% CI -69.50, 39.50, t = -0.623, p = 0.549) (Fig 5) while the effect size was small ($\eta$ 2p = 0.041).

## Discussions

This study investigated the effects of an exercise session during the follicular phase compared with the luteal phase on cortisol, substrate utilization, and cognitive function.

The results of this study showed that there was no significant difference between resting cortisol levels in the follicular and luteal phases. Although the resting cortisol level in the follicular phase was higher(non-significantly) than that in the luteal phase, the effect size was large (18%). Similar to our findings, a previous study found that circulating cortisol levels were higher in the follicular phase than in the luteal phase [11]. A possible reason for our non-significant results may be the small number of participants in this study.

Another finding of this study was that cortisol levels increased during the follicular and luteal phases after exercise compared with before-exercise performance. However, exercise cortisol levels were higher in the follicular phase than in the luteal phase. Sex hormones may interact with cortisol during exercise. During the follicular phase (days 7–10 of the menstrual cycle), estrogen levels are similar to those during the luteal phase (days 21–24 of the menstrual cycle), whereas progesterone levels are lowest in the follicular phase and highest in the luteal phase [13]. Progesterone metabolites positively modulate gamma-aminobutyric acid A (GABA) receptors via an allosteric binding site to enhance inhibitory signaling and increase negative feedback on the hypothalamic-pituitary-adrenal (HPA) axis, leading to a decrease in circulating cortisol [11]. In response to exercise, the HPA axis activities and cortisol levels increase, but these activities can be suppressed by the interaction as mentioned earlier with progesterone in the luteal phase.

Another study results showed that fat utilization during the follicular phase (pre-ovulation phase) was higher than in the luteal phase. Similar to our findings, another study indicated that the peak fat oxidation rate is not increased by natural physiological fluctuations in estrogen. In healthy, young eumenorrheic women, the menstrual cycle phase does not influence maximal fat oxidation intensity during a graded exercise test [27]. In contrast to this study, Willette et al. investigated how estradiol-β-17 during different menstrual cycle phases affects energy substrate utilization and oxidation during aerobic exercise. The study involved physically active women with regular menstrual cycles who participated in two steady-state running

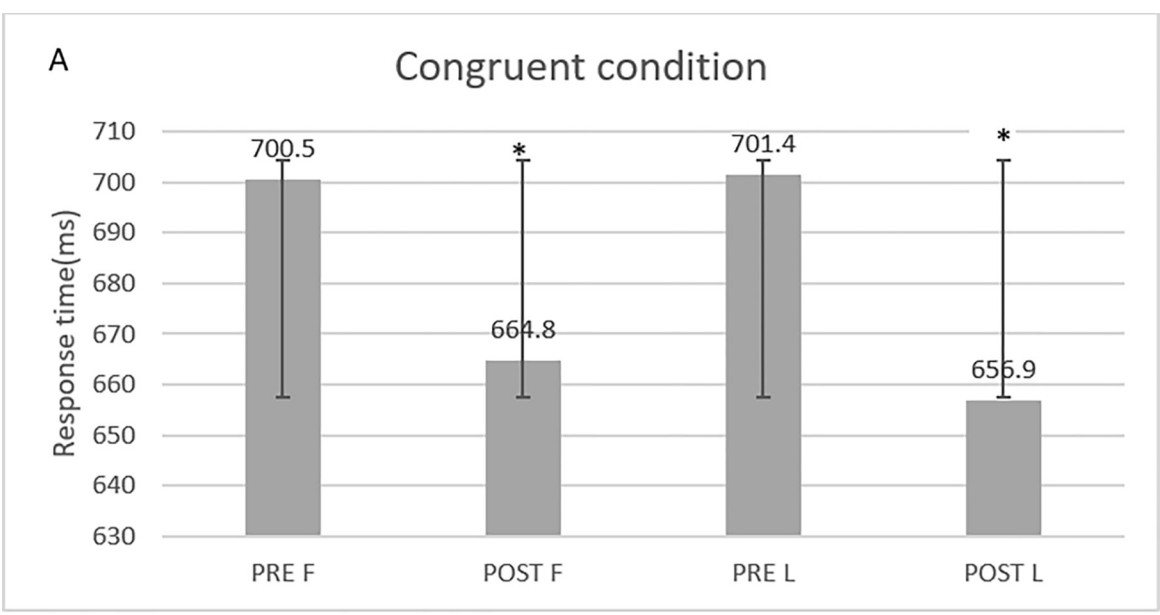

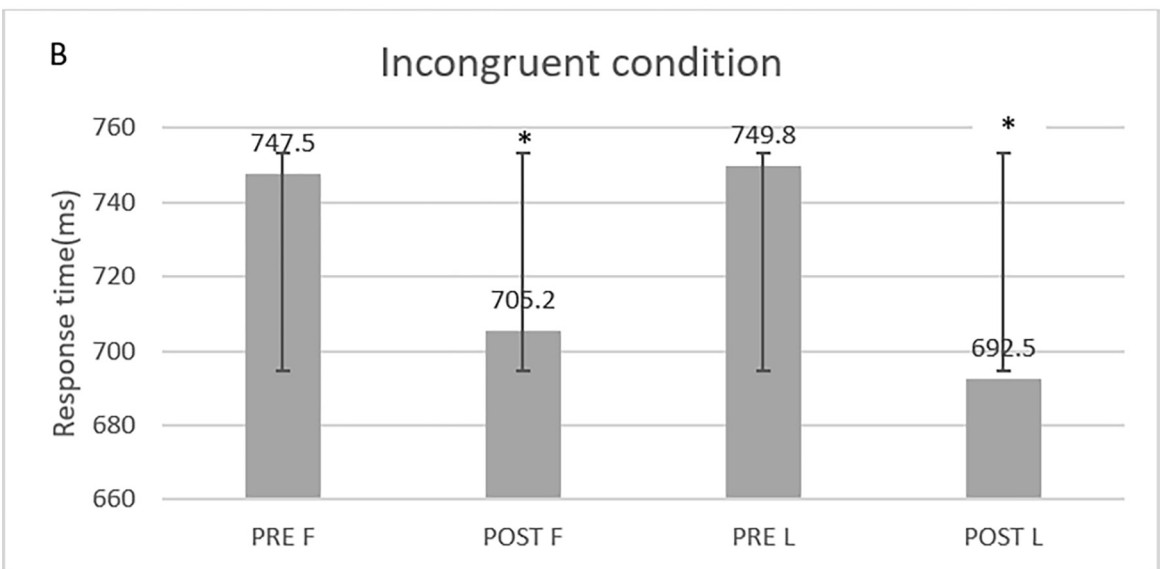

**Fig 5.** Comparison of the reaction time of congruent(A) and incongruent(B) colors in four different times (pre-exercise follicular phase (PRE F), post-exercise follicular phase (POST F), pre-exercise luteal phase (PRE L) and post-exercise luteal phase (POST L)). * significant difference compared with pre-exercise.

sessions at 65% of VO2max, one during the early follicular phase and the other during the luteal phase. The findings indicate a higher utilization of fat and a decreased use of carbohydrates during the luteal phase compared with the follicular phase, which was directly associated with changes in resting estradiol-β-17 [28]. Our different findings may be because the days of the follicular and luteal phases in the present study and the aforementioned study do not coincide, which affects the hormonal status and results.

In addition to higher fat utilization in the follicular phase, our study showed that cortisol levels were higher in the follicular phase after an exercise session. Cortisol promotes lipolysis

or fat utilization through different genomic and non-genomic signaling pathways. Genomic pathways are mediated directly through the cytosolic cortisol receptor, which regulates the expression of proteins required for lipid metabolism, such as adipose triglyceride lipase (ATGL) and hormone-sensitive lipase (HSL). Nongenomic mechanisms are often not dependent on cytosolic cortisol and occur rapidly after cortisol exposure [29]. Increased cortisol levels due to exercise stress stimulate lipolysis and the release of fatty acids as substrates for energy production. However, these effects depend on the exercise type. In this study, lower levels of progesterone during the follicular(pre-ovulation) phase could cause the difference between the two follicular and luteal phases. Progesterone decreases the rate of lipolysis and inhibits ATGL level by stimulating the synthesis of G0S2 –a specific inhibitor of HSL, by inhibiting Lipe gene expression [30].

However, in the present study, RER, an indicator of the ratio of carbohydrate to fat utilization did not differ significantly different between follicular and luteal phases. Different RER responses during exercise could be related to various factors. A diet that enhances systemic alkalinity may increase RER and substrate utilization to favor lipid oxidation and spare carbohydrate utilization [31]. In this study, we did not control the acid-base characteristics of the participants' usual diet which could affect our findings. Additionally, it is important to note that in the present study, RER in the follicular phase (0.892±0.076) was lower(non-significantly) than in the luteal phase (0.893±0.057), which indicated slightly higher lipolysis in the follicular phase. The statistically non-significant finding may also be related to the small number of participants. Energy expenditure during the follicular phase (460±138.755) was also higher(non-significantly) than in the luteal phase (421.30±81.177), which could be related to higher fat utilization during the follicular phase [32]. An increase in energy expenditure can increase the demand for substrate utilization, especially in the form of fat.

Considering cognitive function, this research showed that at rest, there was no significant difference between follicular and luteal reaction time to congruent and incongruent colors. Following exercise, both in the follicular and luteal phases, the reaction times to congruent and incongruent colors were significantly reduced after exercise compared with before exercise. A comparison of the follicular and luteal phases indicated that after exercise, there was no significant difference during the two phases, considering the reaction time to congruent and incongruent colors. In other words, cognitive function improved in response to exercise during both menstrual phases without any significant differences between phases. No study was found regarding the effect of menstrual phases on cognitive function in response to exercise. One study found a positive effect of six months of estrogen therapy on verbal memory in athletes [33]. Consistent with our findings, a study conducted by Wang et al. on collegiate students without considering the phases of the menstrual cycle found that the reaction time to congruent and incongruent colors in the Stroop test decreased after 20 min of exercise on the ergometer bike compared to before exercise [34]. While it was shown that 20–30 min of moderate-intensity cycling (50–60% VO2max) increased cerebral cortex oxygenation after exercise (i.e., activation), which was associated with improved executive performance after exercise (Stroop task). However, contrary to this study, Peltzer et al. observed that regardless of performance, frontostriatal activation increased during the luteal phase when progesterone levels were high [35]. In addition, in another study, in the luteal phase, a decrease in inhibition was observed in the frontostriatal areas related to higher cognitive effort [36].

The findings of this study also indicated that cortisol levels and fat oxidation increased in the follicular phase following exercise. Given the relationship between higher mean cortisol levels [37] and serum fatty acid [12] and cognitive performance, it was expected that cognitive function could improve during the follicular phase. A positive correlation was found between increased cortisol release after exercise and vocabulary retention among healthy adults [38].

However, cognitive function is affected by several physiological factors other than lipolysis or cortisol, and its main result may be a non-significant difference between the follicular and luteal phases. For example, improvements in cognitive function following short-term aerobic exercise may be partially mediated through cerebral blood flow, cerebral cortex activation, growth, and neurotrophic factors, as well as hormones such as NE, dopamine (DA), myokines [39], and cerebral oxygenation [40]. In confirmation of our results, the findings of a review study did not show consistent, clinically important effects of progesterone on cognitive function in women [41], indicating the necessity of more studies for clarification.

Generally, so far, in the studies conducted on women, only the two follicular and luteal periods have been mentioned, and considering the many hormonal changes even during every one of these two periods and the results of the present research, more specifically considering the days of these two periods and considering more physiological mechanisms are recommended.

## Strengths, limitations, and future research

The strength and novelty of this study lie in two main aspects. Firstly, it evaluated the impact of exercise during two distinct phases of the menstrual cycle: the pre-ovulatory phase (days 7–10) and the mid-luteal phase (days 21–24), where estrogen levels are similar, but progesterone levels differ. It also examined the effect of progesterone. Secondly, the study explored the cognitive function response to exercise, considering the potential mediating roles of cortisol and metabolism.

The limitations of this study include the small number of participants, lack of control over the participants' mental conditions before exercise, and failure to measure cortisol levels during a time series in recovery and measuring plasma cortisol levels, as well as not measuring effective hormones of estrogen and progesterone which are recommended for consideration in future studies.

From an applied perspective, the findings can have several implications. Coaches and female athletes can utilize the results of this research in sports where cognitive performance is crucial. This will enable them to compete in the follicular and luteal phases without any worries. It is recommended to exercise during the follicular (pre-ovulation) phase to enhance fat utilization, particularly for individuals aiming to reduce adipose tissue. Given the research's limitations, notably the small number of participants, further studies are needed to generalize its results.

## Conclusions

In summary, a higher fat metabolism was observed during the follicular phase (characterized by a high amount of estrogen and the lowest amount of progesterone) compared to the luteal phase (marked by high amounts of estrogen and progesterone) following exercise. There was a greater exercise-induced increase in cortisol levels in the follicular phase compared to the luteal phase. Cognitive performance after exercise was similar in the follicular and luteal phases. Therefore, other factors affecting cognitive function are suggested to be further elucidated in future studies. Also, the findings are according to measurments of the two cycles. So, its generalization must be with caution and more future clarifications may be required.

From a practical point of view, according to the results of this research, it is recommended to focus more on exercise training during the follicular (pre- ovulation) phase in order to reduce body fat.

## Supporting information

**S1 File. IBM SPSS Statistics raw data file.**
(SAV)

**S2 File. Raw data in excel worksheet.** https://doi.org/10.6084/m9.figshare.25941154.v1.
(XLSX)

## Acknowledgments

The authors appreciate all patients who participated in this study.

## Author Contributions

**Conceptualization:** Mohammad Hemmatinafar, Maryam Koushkie Jahromi.

**Formal analysis:** Maryam Mirzaei Khalil Abadi.

**Funding acquisition:** Maryam Koushkie Jahromi.

**Investigation:** Maryam Mirzaei Khalil Abadi, Mohammad Hemmatinafar.

**Project administration:** Maryam Mirzaei Khalil Abadi.

**Supervision:** Maryam Koushkie Jahromi.

**Writing – original draft:** Maryam Mirzaei Khalil Abadi, Maryam Koushkie Jahromi.

**Writing – review & editing:** Maryam Koushkie Jahromi.

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
