## [Decision Letter · Decision Letter 0]

18 Jul 2024

PONE-D-24-25027Effects of menstrual cycle on cognitive function, cortisol and metabolism after a single session of  aerobic exercisePLOS ONE

Dear Dr. Jahromi,

Thank you for submitting your manuscript to PLOS ONE. After careful consideration, we feel that it has merit but does not fully meet PLOS ONE’s publication criteria as it currently stands. Therefore, we invite you to submit a revised version of the manuscript that addresses the points raised during the review process.

We look forward to receiving your revised manuscript.

Kind regards,

Ayman Abdelaziz Swelum

Academic Editor

PLOS ONE

Journal Requirements:

2. In the online submission form, you indicated that [Data are availbale by corresponding author and will be delivered on request.]. 

Reviewers' comments:

Reviewer's Responses to Questions

**Comments to the Author**

1. Is the manuscript technically sound, and do the data support the conclusions?

Reviewer #1: Partly

Reviewer #2: Yes

Reviewer #3: Yes

2. Has the statistical analysis been performed appropriately and rigorously? 

Reviewer #1: No

Reviewer #2: Yes

Reviewer #3: I Don't Know

3. Have the authors made all data underlying the findings in their manuscript fully available?

Reviewer #1: Yes

Reviewer #2: Yes

Reviewer #3: Yes

4. Is the manuscript presented in an intelligible fashion and written in standard English?

Reviewer #1: No

Reviewer #2: Yes

Reviewer #3: No

5. Review Comments to the Author

Reviewer #1: Thank you for such an interesting study. I have a few comments and suggestions, if you don't mind:

1- In the abstract: would you mind Please to structure the abstract by including opening keywords before each paragraph, such as Aim, Study Design, Results, and Conclusion.

2- In the study materials:

- Study setting?

- Would you mind to highlight the type of the study?

- Has any one of the statistical professionals calculated the sample size for you prior to the study? as 12 sample size could be carrying a limitation of the statistical power and increase type 2 error.

- Could you please put a measurable identifications for the different terms you used in the inclusion criteria such as:noramal regular cycle, no obesity , follicular phase, luteal phase.

- You started your study with 12 participants, however you devided them into 5 and 5 in each phasse, any explanation?

2- Would you mind updating some of your references?

3- Highlighting strength and limitations of your study would give a realistic impression and would reflect positively on the study application.

4- The article would benefit from English proofreading as there are several grammatical mistakes.

Reviewer #2: An interesting and novel study,thank you for the effort done,i may suggest in future research to recruit a bigger sample and give longer period for follow up ,also to correlate the results with clinical outcome like fertility.

Reviewer #3: ABSTRACT: The sentence in lines 6-8 should be recasted and grammar corrected.

INTRODUCTION: Some grammatical errors that need to be corrected. Page 4, paragraph 1, the last sentence is not clear and should be recasted.

METHODS: The authors should state the university where the study was conducted and its location and also how they determined their sample size. Page 4, 1st paragraph, line 3, what do the authors mean by active young female students? The authors should explain how they selected those who participated in the study, what they mean by - physical and mental health to exercise (page 4, last line), how they determined those who were obese in order to exclude them. They should also give examples of the diseases that could affect study variables. The authors should state the name of the ethics review board that approved the conduct of the study.

DISCUSSION: There are grammatical errors that make some sentences difficult to understand. Under strengths and limitations, the last sentence is not clear and should be recasted.

REFERENCES" Some do not appear complete eg numbers 17, 21 and 25

6. PLOS authors have the option to publish the peer review history of their article (what does this mean?). If published, this will include your full peer review and any attached files.

Reviewer #1: No

Reviewer #2: **Yes: **Mohsen M A Abdelhafez

Reviewer #3: No

---

## [Author Response · Author response to Decision Letter 0]

1 Aug 2024

Reviewer Response

Title: Effects of menstrual cycle on cognitive function, cortisol and metabolism after a single session of aerobic exercise

Authors: Maryam Mirzaie Khalilabadi, Mohammad Hemmatinafar & Maryam Koushkie Jahromi

Journal: Plos one We thank the Handling Editor for the opportunity to submit a revised manuscript. We also thank the anonymous Reviewers for their careful and thoughtful comments and critiques.

We believe the reviews helped us to improve the quality of the manuscript. 

Please find the detailed point-by-point-response below.

As recommended, all changes were included in track change. So new pages have been mentioned in revisions.

Of course, we are ready to do more work on the files, in the event that the revision and the answers are not yet entirely satisfactory. 

2. In the online submission form, you indicated that [Data are availbale by corresponding author and will be delivered on request.].

Thank you

1. The manuscript writing was adjusted according to recommendation

2. The only available data is spss data and will be available on the request. 

Reviewer #1: Thank you for such an interesting study. I have a few comments and suggestions, if you don't mind:

 Thank you for your encouraging words, devoting your valuable time to reviewing the article. We also appreciate your very important and valuable points that had a great impact on the improvement of the article.

1- In the abstract: would you mind Please to structure the abstract by including opening keywords before each paragraph, such as Aim, Study Design, Results, and Conclusion.

 Thank you. Abstract was structured according to recommendation.

2- In the study materials:

- Study setting?

- Would you mind to highlight the type of the study?

- Has any one of the statistical professionals calculated the sample size for you prior to the study? as 12 sample size could be carrying a limitation of the statistical power and increase type 2 error.

- Could you please put a measurable identifications for the different terms you used in the inclusion criteria such as:noramal regular cycle, no obesity , follicular phase, luteal phase.

- You started your study with 12 participants, however you devided them into 5 and 5 in each phasse, any explanation?

 Thank you. The following sentences or modifications were included:

- The study setting was Shiraz University which is one of the largest and most important universities in the country, and more than 20000 students from different cities of Iran.

- This was a nonrandomized crossover study.

- That’s right. Unfortunately we tried as much as possible. However, we could not find any more subject. A method of estimating sample size using G*power was included. 

- 

- In addition, the small number of participants was included as limitation of the study. 

- Normal regular cycle, obesity follicular and luteal phases were defined . 

- Sorry for mistake. It was added. (Introduction and method were provided before the final procedure of experiment and 10 girls predicted to participate while 12 persons participated in final stage. 

- Would you mind updating some of your references? Thank you I tried to update references as much as possible and hope it is acceptable.

3- Highlighting strength and limitations of your study would give a realistic impression and would reflect positively on the study application.

 Thank you . I tried to complete limitations and applications. 

4- The article would benefit from English proofreading as there are several grammatical mistakes.

 Thank you. I tried to modify the English writing.

Reviewer #2: An interesting and novel study,thank you for the effort done,

I may suggest in future research to recruit a bigger sample and give longer period for follow up ,also to correlate the results with clinical outcome like fertility.

 We greatly appreciate your positive and encouraging words.

Thank you for your important suggestion. 

- That’s right. Unfortunately we tried as much as possible. However, we could not find any more subject. A method of estimating sample size using G*power was included. 

I mentioned them as limitations of the study and surely will use them in my future research. 

Reviewer #3: 

ABSTRACT: The sentence in lines 6-8 should be recasted and grammar corrected.

 We greatly appreciate all your suggestions and important points.

Thank you. Abstract was modified. 

INTRODUCTION: Some grammatical errors that need to be corrected. Page 4, paragraph 1, the last sentence is not clear and should be recasted.

 Thank you. Grammar was checked in all parts of manuscript and modifications were considered. 

METHODS: The authors should state the university where the study was conducted and its location

 and also how they determined their sample size. 

Page 4, 1st paragraph, line 3, what do the authors mean by active young female students? 

The authors should explain how they selected those who participated in the study,

 what they mean by - physical and mental health to exercise (page 4, last line), 

how they determined those who were obese in order to exclude them. 

They should also give examples of the diseases that could affect study variables. 

The authors should state the name of the ethics review board that approved the conduct of the study. - Thank you

- The university name was added.

- That’s right. Unfortunately we tried as much as possible. However, we could not find any more subject. A method of estimating sample size using G*power was included. 

- Active women means persons who had a history of participating in sports training regularly at least two sessions a week in the past year.(this sentence was added).

- Participants were recruited through announcements in university dormitories and gymnasiums(sentence was added)

- 

- Mental and physical health was approved by a physician

- 

- The criteria for obesity was mentioned

- Examples of disease that could affect the study were added. 

_ the name of ethic committee was added. 

DISCUSSION: There are grammatical errors that make some sentences difficult to understand.

 Under strengths and limitations, the last sentence is not clear and should be recasted.

 Thank you_grammar was checked in all manuscript and errors were modified. 

_ the last sentence was modified . 

REFERENCES" Some do not appear complete eg numbers 17, 21 and 25 Thanks References were completed.

 Thanks I tried several times and no change was made on figures by PACE so I submitted in previous format.

If I need to do anything else I greatly appreciate let me know.

---

## [Decision Letter · Decision Letter 1]

8 Sep 2024

PONE-D-24-25027R1Effects of menstrual cycle on cognitive function, cortisol, and metabolism after a single session of  aerobic exercisePLOS ONE

Dear Dr. Koushkie Jahromi, Thank you for submitting your manuscript to PLOS ONE. After careful consideration, we feel that it has merit but does not fully meet PLOS ONE’s publication criteria as it currently stands. Therefore, we invite you to submit a revised version of the manuscript that addresses the points raised during the review process.

**ACADEMIC EDITOR: ****The authors have improved the manuscript in its revised version. However, a revision is mandatory before its acceptance. Please follow all the reviewers comments and reply to their questions. **==============================

We look forward to receiving your revised manuscript.

Kind regards,

Ayman A. Swelum

Academic Editor

PLOS ONE

Reviewers' comments:

Reviewer's Responses to Questions

**Comments to the Author**

1. If the authors have adequately addressed your comments raised in a previous round of review and you feel that this manuscript is now acceptable for publication, you may indicate that here to bypass the “Comments to the Author” section, enter your conflict of interest statement in the “Confidential to Editor” section, and submit your "Accept" recommendation.

Reviewer #3: All comments have been addressed

Reviewer #4: (No Response)

Reviewer #5: (No Response)

Reviewer #6: (No Response)

2. Is the manuscript technically sound, and do the data support the conclusions?

Reviewer #3: Yes

Reviewer #4: Partly

Reviewer #5: Yes

Reviewer #6: Partly

3. Has the statistical analysis been performed appropriately and rigorously? 

Reviewer #3: Yes

Reviewer #4: Yes

Reviewer #5: Yes

Reviewer #6: Yes

4. Have the authors made all data underlying the findings in their manuscript fully available?

Reviewer #3: Yes

Reviewer #4: No

Reviewer #5: Yes

Reviewer #6: Yes

5. Is the manuscript presented in an intelligible fashion and written in standard English?

Reviewer #3: (No Response)

Reviewer #4: Yes

Reviewer #5: Yes

Reviewer #6: Yes

6. Review Comments to the Author

Reviewer #3: (No Response)

Reviewer #4: Manuscript Number: PONE-D-24-25027R1

Article Type: Research Article

Full Title: Effects of menstrual cycle on cognitive function, cortisol, and metabolism after a single session of aerobic exercise

Overall, Thanks for an interesting study. However, this reviewer found some issues and concerns in this manuscript.

1. Authors should maximize the accessibility and reusability of the data by selecting a file format from which data can be efficiently extracted (for example, spreadsheet are preferable to PDFs or images when providing tabulated data). Please see Plos Data in Supporting Information files on https://journals.plos.org/plosone/s/data-availability#loc-acceptable-data-sharing-methods

2. Authors may use a description of supporting information. Please see https://journals.plos.org/plosone/s/supporting-information

3. Abstract: Conclusion:

It is better to add the word “ may” before (do not affect cognitive function). Or replace the follicular and luteal phases with “days 7–10 (follicular) and days 21-24 (mid-luteal phase) do not occur… etc.”

4. Intoduction:

Please mention the physiological importance of days 7–10 (follicular) and days 21-24 (mid-luteal phase) in the menstrual cycle and its importance on cognitive function (in comparison to other studies).

5. Material and Methods:

I. The Experiments must have appropriate sample sizes as 12 sample size carrying a limitation of the statistical power as may reduce the confirmability of the results.

II. What is the material status of the participants and if they nulliparous, primiparous, or multiparous? should be mentioned and in the abstract too.

III. The study has no hormonal analysis or clinical observation to confirm the days of menstrual and other physiological parameters/conditions.

IV. The suitability of the crossover method in this study needs to be clarified. Why did you choose it?

V. The study relied on salivary cortisol measurements, and there was no measurement of plasma cortisol levels, which might provide a more accurate assessment of systemic cortisol responses to exercise.

6. Discussion:

The study has no hormonal analysis indicating the estrogen/progesterone level. Therefore, it would be better to put a reference in “During the follicular phase (days 7–10 of the menstrual cycle), estrogen levels are similar to those during the luteal phase (days 21–24 of the menstrual cycle)”.

Reviewer #5: Well, that was an interesting one, thank you very much to raise such an interesting subject.

I have the following comments:

(1)in the exclusion criteria:

(a)can you give us a simple ideas about the number of volunteers who were excluded due to any of the criteria ?

for example those with HADS score > 7.

(b) why you have limited the included age 22 to 30, why not more , for example 18 to 35, that is 22 seems to be the age of graduation.

(2)in the "Procedure" Section:

(a)have you used any means to determine the date of ovulation ? so that you can determine precisely the date of exercise.

(b)have you confirmed the ovulation ? since in cases of unovulation, you will not have the required level of progesterone in the leuteal phase to exert its physiological effects.

(c)since the core theoretical background of the study is the effect of estradiol and counter effect of progesterone, I think it was prudent to measure both in either phases of the cycle, besides, leuteal phase progesterone level will give us a hint about ovulation, and here, may be, we can find a mathematical correlation between the hormonal levels and the factors studied.

(d)did you give instructions about smoking prior to exercise (as it was given with regards to alcohol), this need to be mentioned in this section.

(3)in the "Exercise protocol" Section:

the first paragraph was repeated.

(4)in the "Saliva sampling and assay" Section:

I feel the ambiguity of the biochemical analysis method, can you make it more clear ?.

(5)in the "Cognitive function" Section:

in the text you need to mention the time unit (ms) at least once.

(6)in the last paragraph of the results:

"In addition, comparison of changes (post-exercise – pre-exercise) using t-test indicated that reaction time for congruent colors was not statistically significant"

I think it is "incongruent", not "congruent", please check that.

(7)in the "Discussion" section:

(a)in the second paragraph you have mentioned "HPA" as an abbreviation in its first occurrence in the text of article, please to be preceded by its full name: "hypothalamic-pituitary-adrenal axis" at this site.

(b)you have mentioned the following:

"RER in the follicular phase (0.892±0.076) was lower(non-significantly) than in the luteal phase (0.893±0.057), which indicated higher lipolysis in the follicular phase"

I think the reverse is true, please cross check with reference cited.

Thank you and I was really interesting reading and reviewing the article.

Reviewer #6: Dear authors, thank you for your interesting research and focus on performance of women athletes and exercise in women in general. I see that important changes have been made regarding the previous review, so this is greatly appreciated. The article is read more easily and the structure is more visible and clear. The statistical analysis is very well described and performed and the data well presented. I would like to address some points I consider important for the interpretation of the study and for creating further research:

- I guess no treatments were taken by the participants, nor suplements that can affect menstrual cycle. Since it is not mentioned and it is the base of the study, I suggest to indicate it. One to mention is the absence of hormonal treatment (e.g. oral contraceptives).

- I would focus more on describing menstrual cycle characteristics since it is the basis of your study: there is no data regarding menstrual cycle duration of the participants. The normal range defined at your study is very wide, and such differences in your participants may affect interpretation of the study (specially in the luteal phase, which can vary greatly).

I guess that ovulation was not assessed and, if so, more precise limitation should be added (e.g. variability in menstrual cycles and, thus, interpretation and comparability of data).

- Since every participant got measured in 2 cycles, did you compare data between first and second cycle? Some differences might be used as extra-information if you find them useful or limitation in case relevant differences are observed (because of variation in the same woman accross cycles).

- As other reviewers have appointed, the limitations should be added in depth: the results of the study are very interesting and they can make a difference for future studies, but conclusions cannot be made based on just 2 cycles: usually, implications are seen in the long term (such as cognition and modification of the amount of fat tissue) and many factors may influence menstrual cycle and thus, training.

- I would recommend as well to point out some strenghts of your study if you might consider: the type of information you want to provide women with, the type of training considering the phase of the cycle, and the importance of knowledge of self cycles and feelings in training to improve the response to exercise and, in general, including menstrual health when considering about all aspects of life and, such as this study, in exercising.

I would suggest extra review of punctuation to read it even more easily after the great corrections already made.

I hope you find the review useful and thank you again for your work.

7. PLOS authors have the option to publish the peer review history of their article (what does this mean?). If published, this will include your full peer review and any attached files.

Reviewer #3: No

Reviewer #4: No

Reviewer #5: **Yes: **Nassar Taha Yaseen Alibrahim

Reviewer #6: No

---

## [Author Response · Author response to Decision Letter 1]

25 Sep 2024

PONE-D-24-25027R1

Effects of menstrual cycle on cognitive function, cortisol, and metabolism after a single session of aerobic exercise

PLOS ONE

ACADEMIC EDITOR: 

The authors have improved the manuscript in its revised version. However, a revision is mandatory before its acceptance. Please follow all the reviewers comments and reply to their questions. 

We thank the Academic Editor, Dr Prof Ayman A. Swelum for giving us the opportunity to submit a revised manuscript. We also thank the anonymous Reviewers for their careful and thoughtful comments and critiques.

We believe the reviews helped us to improve the quality of the manuscript. 

Please find the detailed point-by-point-response below.

As recommended, all changes were included in track change. So new pages have been mentioned in revisions.

Of course, we are ready to do more work on the files, in the event that the revision and the answers are not yet entirely satisfactory.

6. Review Comments to the Author

Overall, Thanks for an interesting study. However, this reviewer found some issues and concerns in this manuscript.

We greatly appreciate your positive feedback, all guidance, points and efforts which lead to improving the quality of our manuscript. 

1. Authors should maximize the accessibility and reusability of the data by selecting a file format from which data can be efficiently extracted (for example, spreadsheet are preferable to PDFs or images when providing tabulated data). Please see Plos Data in Supporting Information files on https://journals.plos.org/plosone/s/data-availability#loc-acceptable-data-sharing-methods

I have provided data in Excel format and has been stored by doi: 10.6084/m9.figshare.25941154. No other direct data is available.

Supporting information is also available through the following link:

https://doi.org/10.6084/m9.figshare.25941154.v1

2. Authors may use a description of supporting information. Please see https://journals.plos.org/plosone/s/supporting-information

There is no need for supporting information. I have just spss file of data. 

3. Abstract: Conclusion:

It is better to add the word “ may” before (do not affect cognitive function).

Thank you. May was added in place of do.

 Or replace the follicular and luteal phases with “days 7–10 (follicular) and days 21-24 (mid-luteal phase) do not occur… etc.”

In abstract follicular and luteal phases were replaced by “days 7–10 (follicular) and days 21-24 (mid-luteal phase)

4. Introduction:

Please mention the physiological importance of days 7–10 (follicular) and days 21-24 (mid-luteal phase) in the menstrual cycle and its importance on cognitive function (in comparison to other studies).

Thank you. The physiological mechanisms were compared and added during two phases in introduction. 

5. Material and Methods:

I. The Experiments must have appropriate sample sizes as 12 sample size carrying a limitation of the statistical power as may reduce the confirmability of the results.

Unfortunately we could not accommodate for more participants. It was included as a limitation of this study. 

II. What is the material status of the participants and if they nulliparous, primiparous, or multiparous? 

Thank you. This sentence was added: All of them were unmarried and had no history of pregnancy or childbirth. 

III. The study has no hormonal analysis or clinical observation to confirm the days of menstrual and other physiological parameters/conditions.

Menstrual cycle days were estimated according to the following method which was available and completed in the manuscript:

To monitor the regularity and date of menstruation, participants were taught to record the exact date of the start and end of their menstrual cycle and bleeding, and their resting heart rate (after waking) using a Polar H10 Heart Rate Sensor(made in Kempele, Finland) for three months before the start of the study. The validity of estimating ovulation through heart rate has been approved [21]. A slight increase in resting heart rate as well as recorded information considering menustrual cycle were considered for predicting follicular, luteal, and ovulation dates, The follicular (beginning from the first day of menses until ovulation) and luteal (beginning around day 15 of a 28-day cycle and ending with menstruation) phases of the participants were estimated and recorded by the researcher according to the information provided.

IV. The suitability of the crossover method in this study needs to be clarified. Why did you choose it?

Thank you. Cross over design was used to limit the effect of learning and adaptation(it was added).

V. The study relied on salivary cortisol measurements, and there was no measurement of plasma cortisol levels, which might provide a more accurate assessment of systemic cortisol responses to exercise.

You are right, plasma cortisol may be more accurate , however there is a strong correlation between plasma and salivary cortisol according to various studies(e.g. Thomasson R, Baillot A, Jollin L, Lecoq AM, Amiot V, Lasne F, Collomp K. Correlation between plasma and saliva adrenocortical hormones in response to submaximal exercise. J Physiol Sci. 2010 Nov;60(6):435-9.) Because we required 4 measurements of cortisol, we preferred to measure noninvasively and through saliva measure cortisol. 

6. Discussion:

The study has no hormonal analysis indicating the estrogen/progesterone level. Therefore, it would be better to put a reference in “During the follicular phase (days 7–10 of the menstrual cycle), estrogen levels are similar to those during the luteal phase (days 21–24 of the menstrual cycle)”.

Thank you. The reference was inserted. 

Reviewer #5: Well, that was an interesting one, thank you very much to raise such an interesting subject.

I have the following comments:

We greatly appreciate your positive feedback, all guidance, points and efforts which lead to improving the quality of my manuscript. 

(1)in the exclusion criteria:

(a)can you give us a simple ideas about the number of volunteers who were excluded due to any of the criteria ?

for example those with HADS score > 7.

Thank you , the following sentence was added.

Among the 26 volunteers participating in the study, 15 were selected according to the inclusion criteria and three were excluded according to the HAD score.

(b) why you have limited the included age 22 to 30, why not more, for example 18 to 35, that is 22 seems to be the age of graduation.

Participants were selected from post graduate students who were at the age range of 22-30, because they did not have physical education course and their daily activity was more stable than undergraduate students. Also, their cooperation in study programs and availability was much better due to understanding the necessity of their cooperation. 

(2)in the "Procedure" Section:

(a)have you used any means to determine the date of ovulation ? so that you can determine precisely the date of exercise.

There was some information in this regard which was compeleted as follow:

To monitor the regularity and date of menstruation, participants were taught to record the exact date of the start and end of their menstrual cycle and bleeding, and their resting heart rate (after waking) using a Polar H10 Heart Rate Sensor (made in Kempele, Finland) for three months before the start of the study. The validity of estimating ovulation through heart rate has been approved [21]. A slight increase in resting heart rate as well as recorded information considering menustrual cycle were considered for predicting follicular, luteal, and ovulation dates, The follicular (beginning from the first day of menses until ovulation) and luteal (beginning around day 15 of a 28-day cycle and ending with menstruation) phases of the participants were estimated and recorded by the researcher according to the information provided.

(b)have you confirmed the ovulation ? since in cases of unovulation, you will not have the required level of progesterone in the leuteal phase to exert its physiological effects.

Yes, during these three months we confirmed ovulation through the method mentioned in the previous question.

(c)since the core theoretical background of the study is the effect of estradiol and counter effect of progesterone, I think it was prudent to measure both in either phases of the cycle, besides, leuteal phase progesterone level will give us a hint about ovulation, and here, may be, we can find a mathematical correlation between the hormonal levels and the factors studied.

Thank you. You are completely right, it was the best if we could measure these two hormones, however because we required to take several blood samples and participants were not satisfied with this matter and their salivary measurements were too expensive, we could not measure hormones and we mentioned it as a limitation of this study.

(d)did you give instructions about smoking prior to exercise (as it was given with regards to alcohol), this need to be mentioned in this section.

Thank you, it was added(although in our culture smoking and drinking alcohol is very unusual and bad matter for educated persons specially girls, however it was added)

(3)in the "Exercise protocol" Section:

the first paragraph was repeated.

Thank you. It was deleted/ 

(4)in the "Saliva sampling and assay" Section:

I feel the ambiguity of the biochemical analysis method, can you make it more clear ?.

Thank you. I tried to make some clarifications. 

(5)in the "Cognitive function" Section:

in the text you need to mention the time unit (ms) at least once.

The following sentence was added:

the speed of their response was recorded in milliseconds(ms).

(6)in the last paragraph of the results:

"In addition, comparison of changes (post-exercise – pre-exercise) using t-test indicated that reaction time for congruent colors was not statistically significant"

I think it is "incongruent", not "congruent", please check that.

Thank you. It was modified. 

(7)in the "Discussion" section:

(a)in the second paragraph you have mentioned "HPA" as an abbreviation in its first occurrence in the text of article, please to be preceded by its full name: "hypothalamic-pituitary-adrenal axis" at this site.

Thank you for your careful considerations.

It was added .

(b)you have mentioned the following:

"RER in the follicular phase (0.892±0.076) was lower(non-significantly) than in the luteal phase (0.893±0.057), which indicated higher lipolysis in the follicular phase"

I think the reverse is true, please cross check with reference cited.

Thank you. Lower RER is associated with higher lipolysis. In two phases that were measured ((pre-ovulation: days 7–10) and luteal (mid-luteal: days 21-24) the amount of estrogen is similar, while in some other previous studies other days of menstruation may have been considered. This may have caused different findings. Also, our finding was statistically non-significant. So, we could not strongly express our findings.

Its explanation and reference were modified for better clarifications. 

Thank you and I was really interesting reading and reviewing the article.

I greatly appreciate your positive feedback and all your efforts. 

Reviewer #6: Dear authors, thank you for your interesting research and focus on performance of women athletes and exercise in women in general. I see that important changes have been made regarding the previous review, so this is greatly appreciated. The article is read more easily and the structure is more visible and clear. The statistical analysis is very well described and performed and the data well presented. I would like to address some points I consider important for the interpretation of the study and for creating further research:

We greatly appreciate your positive feedback, all guidance, points and efforts which lead to improving the quality of my manuscript. 

- I guess no treatments were taken by the participants, nor suplements that can affect menstrual cycle. Since it is not mentioned and it is the base of the study, I suggest to indicate it. One to mention is the absence of hormonal treatment (e.g. oral contraceptives).

That is right no treatment , no supplement and no intention was performed to affect manse. Thank you, Absence of hormonal treatment was added as inclusion criteria, 

- I would focus more on describing menstrual cycle characteristics since it is the basis of your study: there is no data regarding menstrual cycle duration of the participants. The normal range defined at your study is very wide, and such differences in your participants may affect interpretation of the study (specially in the luteal phase, which can vary greatly).

Thank you . menstrual cycle duration was 27-30 days which was added in results. 

I guess that ovulation was not assessed and, if so, more precise limitation should be added (e.g. variability in menstrual cycles and, thus, interpretation and comparability of data).

The method of estimating ovulation was completed as follow. However, the menstrual cycle mean and SD was added in results. To monitor the regularity and date of menstruation, participants were taught to record the exact date of the start and end of their menstrual cycle and bleeding, and their resting heart rate (after waking) using a Polar H10 Heart Rate Sensor(made in Kempele, Finland) for three months before the start of the study. The validity of estimating ovulation through heart rate has been approved [21]. A slight increase in resting heart rate as well as recorded information considering menustrual cycle were considered for predicting follicular, luteal, and ovulation dates, The follicular (beginning from the first day of menses until ovulation) and luteal (beginning around day 15 of a 28-day cycle and ending with menstruation) phases of the participants were estimated and recorded by the researcher according to the information provided.

- Since every participant got measured in 2 cycles, did you compare data between first and second cycle? Some differences might be used as extra-information if you find them useful or limitation in case relevant differences are observed (because of variation in the same woman accross cycles).

Yes, we compared between two cycles . thank you, I mentioned it in the limitation. 

- As other reviewers have appointed, the limitations should be added in depth: the results of the study are very interesting and they can make a difference for future studies, but conclusions cannot be made based on just 2 cycles: usually, implications are seen in the long term (such as cognition and modification of the amount of fat tissue) and many factors may influence menstrual cycle and thus, training.

Thank you . it was mentioned in conclusion. 

- I would recommend as well to point out some strenghts of your study if you might consider: the type of information you want to provide women with, the type of training considering the phase of the cycle, and the importance of knowledge of self cycles and feelings in training to improve the response to exercise and, in general, including menstrual health when considering about all aspects of life and, such as this study, in exercising.

Thank you. I tried to include in the final parts of the manuscript. 

I would suggest extra review of punctuation to read it even more easily after the great corrections already made.

Thank you. The whole manuscript punctuation and grammar were reviewed and modified. 

I hope you find the review useful and thank you again for your work.

Thank you so much. It was really important and helpful. 

---

## [Editor Report · Decision Letter 2]

29 Sep 2024

Effects of menstrual cycle on cognitive function, cortisol, and metabolism after a single session of  aerobic exercise

PONE-D-24-25027R2

Dear Dr. Maryam Koushkie Jahromi,

We’re pleased to inform you that your manuscript has been judged scientifically suitable for publication and will be formally accepted for publication once it meets all outstanding technical requirements.

Kind regards,

Ayman A. Swelum

Academic Editor

PLOS ONE

---

## [Editor Report · Acceptance letter]

18 Oct 2024

PONE-D-24-25027R2 

PLOS ONE

Dear Dr. Koushkie Jahromi, 

I'm pleased to inform you that your manuscript has been deemed suitable for publication in PLOS ONE. Congratulations! Your manuscript is now being handed over to our production team.

Kind regards, 

on behalf of

Professor Ayman A. Swelum 

Academic Editor

PLOS ONE